

# Postharvest biochemical characteristics and ultrastructure of *Coprinus comatus*

Yi Peng[1,2], Tongling Li[2], Huaming Jiang[3], Yunfu Gu[1], Qiang Chen[1], Cairong Yang[2,4], Wei liang Qi[2], Song-qing Liu[2,4] and Xiaoping Zhang[1]

[1] College of Resources, Sichuan Agricultural Uniersity, Chengdu, Sichuan, China
[2] College of Chemistry and Life Sciences, Chengdu Normal University, Chengdu, Sichuan, China
[3] Sichuan Vocational and Technical College, Suining, Sichuan, China
[4] Institute of Microbiology, Chengdu Normal University, Chengdu, Sichuan, China

## ABSTRACT

**Background.** *Coprinus comatus* is a novel cultivated edible fungus, hailed as a new preeminent breed of mushroom. However, *C. comatus* is difficult to keep fresh at room temperature after harvest due to high respiration, browning, self-dissolve and lack of physical protection.

**Methods.** In order to extend the shelf life of *C. comatus* and reduce its loss in storage, changes in quality, biochemical content, cell wall metabolism and ultrastructure of *C. comatus* (*C.c*77) under 4 °C and 90% RH storage regimes were investigated in this study.

**Results.** The results showed that: (1) After 10 days of storage, mushrooms appeared acutely browning, cap opening and flowing black juice, rendering the mushrooms commercially unacceptable. (2) The activity of SOD, CAT, POD gradually increased, peaked at the day 10, up to 31.62 U g$^{-1}$ FW, 16.51 U g$^{-1}$ FW, 0.33 U g$^{-1}$ FW, respectively. High SOD, CAT, POD activity could be beneficial in protecting cells from ROS-induced injuries, alleviating lipid peroxidation and stabilizing membrane integrity. (3) The activities of chitinase, β-1,3-glucanase were significantly increased. Higher degrees of cell wall degradation observed during storage might be due to those enzymes' high activities. (4) The fresh *C. comatus* had dense tissue and every single cell had the number of intracellular organelles which structure can be observed clearly. After 10 d storage, the number of intracellular organelles was declined and the structure was fuzzy, the nucleus disappeared. After 20 d storage, *C. comatus*'s organization was completely lost, many cells were stacked together and the cell wall was badly damaged.

## INTRODUCTION

*Coprinus comatus*, also called as shaggy ink cap, is a novel cultivated edible fungus with significantly commercial potential, hailed as a new preeminent breed of mushroom (*Bo et al., 2010*). Moreover, it is also a delicious and highly nutritious edible fungus, with numerous valuable medicinal compounds (*Fan et al., 2006*), including various bioactive functions, such as hypoglycemic, hypolipidemic, antibacterial effects (*Bailey et al., 1985*), immunomodulation, antitumor (*Jiang et al., 2013*) and antioxidant potential (*Tsai, Tsai*

Corresponding authors
Song-qing Liu, biosq@126.com
Xiaoping Zhang,
zhangxiaopingphd@126.com

& *Mau, 2009*). However, *C. comatus* is difficult to keep fresh at room temperature after harvest due to high respiration, browning, self-dissolve and lack of physical protection (*Rui & Feng-Lan, 2006*). The extremely active metabolic process and rapid deterioration of quality have seriously affected the commodity quality and shelf life of *C. comatus*, caused great losses to production and storage.

Under normal growing conditions, the production and clearance of intracellular free radicals and reactive oxygen species (ROS) are in a dynamic balance; however, this balance is destroyed with prolonged storage because the storage conditions are a sort of abiotic stress (*Huang et al., 2007*; *Gill & Tuteja, 2010*), followed by high generation capacity of ROS and low enzymatic antioxidant defense. Superoxide dismutase (SOD), peroxidase (POD), and catalase (CAT) were important parts of enzymatic antioxidant defense system, which can efficiently eliminate ROS (*Duan et al., 2011*). SOD is one of the key enzymes for scavenging free radicals in mushroom, which can transform superoxide anion radicals $(O_2^{\bullet-})$ to $H_2O_2$ and non-toxic molecular oxygen by catalyzing the disproportionation reaction. The CAT and POD could transform $H_2O_2$ into $H_2O$ and $O_2$ through different actions, all of which are crucial for ROS detoxification (*Prasad, Rosangkima & Kharbangar, 2009*).

During postharvest storage of edible fungi, changes in cell wall structure and composition directly contribute to cell separation, resulting in loose organizational structure and decreased hardness of fruiting body and, finally, decreased storability of edible fungi (*Zivanovic, Buescher & Kim, 2003*; *Qi et al., 2015*). Studies have addressed that the content of chitin decreased during postharvest storage due to the effect of chitinase by hydrolyzing the β-1, 4-glycoside bond in chitin and producing n-acetylglucosamine oligomer or monomer (*Buitimea et al., 2013*; *Jiang et al., 2010a*; *Jiang et al., 2010b*). *Lim & Choi (2009)* revealed that the *Agaricus bisporus* could rapidly produce black juice at 25 °C for 15 h after harvest and confirmed that the amount of chitinase synthesis in the autolysis process was significantly higher than that of unpicked mushrooms. *Xue (2012)* studied the effect of postharvest calcium treatment on cell stability of *Lentinula edodes* and found that the stability of cell wall was correlated with β-1, 3-glucan and β-1, 3-glucanase, by inhibiting the activity of β-1, 3-glucanase, the aging of *L. edodes* could be delayed. *Liu et al. (2015)* suggests that β-1,3-glucanases plays a major role in the autolysis of *Coprinopsis* fruiting bodies. However, the accurate role of these enzymes in altering the cell wall of *C. comatus* after harvest were little known.

The degradation of cell wall substances and membrane lipid peroxidation led to changes in the ultrastructure of cells. Ultrastructure is also called submicroscopic structure. Electron microscope can well show the changes of tissue structure, cell structure, organelle function and differentiation (*Hu, Chen & Xu, 2015*). By observing the ultrastructure of *C. comatus*, we can better understand the degradation mechanism of *C. comatus* after harvest. However, there was no report on the ultrastructure of *C. comatus*.

Therefore, it is necessary to conduct in-depth and systematic research on the physiological and biochemical changes of *C. comatus* after harvesting and try to find influencing factors and possible degradation mechanism of texture deterioration. The objective of this work was to investigate sensory characteristics, chemical properties,

functional components, cell wall metabolic enzymes activity and ultrastructure of *C. comatus* during postharvest storage.

## MATERIALS & METHODS

### Materials

*C. comatus* used in this study were harvested from an edible fungus planting base of Chengdu normal university in Sichuan, China (no. *C.c* 77). *C. comatus* were transported to the laboratory in one hour after picking. Then, they ($60 \pm 5$ g) were screened for uniform size and maturity and absence of mechanical damage, packaged in low density polyethylene sealing bags (0.04 mm thickness, size 18 cm $\times$ 20 cm). After that, samples were stored at 4 °C and 90% relative humidity (RH) for 20 d. Subsequently every 5 days, three replicates were randomly selected and analyzed for each biochemical characteristic.

### Browning analysis

We used a NR110 (Shenzhen 3nh technology co. LTD) type color difference meter (L, a, b) to determine the chromatism of *C. comatus*, and evaluated $\Delta E$, $\Delta E = \left[(L-97)^2 + (a-(-2))^2 + b^2\right]^{1/2}$. L represents luminosity, which corresponds to brightness, a represents the range from magenta to green, b represents the range from yellow to blue. $\Delta E$ represents the overall color difference compared with the ideal or target color of the mushroom ($L = 97$, $a = -2$, $b = 0$). The larger the L, the smaller the $\Delta E$, indicating that the color of *C. comatus* is white and the degree of browning is lower.

### Weight loss analysis

According to the method of *Jiang, Feng & Li (2012)*. It was expressed as the percentage of loss of weight with respect to the initial weight.

### Analysis of membrane permeability

According to the method of *Liu et al. (2010)*. Randomly selected 3–4 *C. comatus*, used a five mm diameter puncher to punch holes in different parts, mixed and sampled 2 g in 50 ml beaker, washed with no ion water several times, then add 30 mL without ionized water to soak 1 h, conductivity of the surrounding solution was determined with a conductivity meter. The tissue was then boiled for 10 min and the total conductivity was recorded. Electrolyte leakage was expressed as a percentage of total electrolytes in the tissue.

### Analysis of chemical properties

Soluble protein was determined according to the method of *Bradford (1976)* using bovine serum albumin as standard. Malondialdehyde (MDA) content was determined according to the method of Jayakumar (*Jayakumar, Ramesh & Geraldine, 2006*). 2 g of the fruit body of *C. comatus* was ground in five mL 5%TCA using a mortar and pestle, and quickly cooled in an ice bath and centrifuged at 8,000 r/min for 10 min. Then two mL supernatant was ground in two mL 0.67%TBA, after heating at 100 °C for 30 min, the mixture was centrifuged again. The absorbance of the supernatant was read at 450 nm, 532 nm and 600 nm respectively. Concentration of MDA: C($\mu$mol/L) = 6.45(A532-A600)-0.56 A450.

$O_2^{\bullet-}$ was determined according to the methods of *Jiang et al. (2010a)* and *Jiang et al. (2010b)*. Weighed 2 g of the mushroom, grinded with six mL of 65 mmol/L (pH 7.8) PBS, centrifuged the filtrate at 7,000 r/min for 5 min at 4 °C, and took the supernatant. Determination of $O_2^{\bullet-}$ content: one mL of the supernatant was taken, and 0.9 mL (pH 7.8) of PBS, 0.1 mL 10 mmol/L of hydroxylamine hydrochloride (distilled water instead of the sample supernatant was used as a blank) was added. After mixing, it was incubated at 25 °C for 20 min. Took 0.5 mL of the above culture solution, added 0.5 mL 17 mmol/L p-aminobenzenesulfonic acid, 0.5 mL seven mmol/L a-naphthylamine, and kept the reaction in a constant temperature water bath at 25 °C for 20 min, added 1.5 mL of n-butanol, after homogenization, the n-butanol phase was taken to determine the OD value at 530 nm. Using $NaNO_2$ as the standard, according to the measured OD530, check the $NO_2^-$ standard curve, convert OD530 into $[NO_2^-]$, and $[NO_2^-] \times 2$ to obtain $[O_2^{\bullet-}]$ content.

## Analysis of functional components

Total phenolics: mushroom tissues (2 g) were homogenized with 80 mL water, boiled at 100 °C for 30 min, settled to permit after cooling, then filtered. Color was developed by mixing 1ml filtrate, five mL water, one mL Folin-Ciocalteu (Sigma–Aldrich Chemical Co., St. Louis, Mo, USA) and 3 ml 7.5% sodium carbonate solution. After 2 h, the absorbance of solution was measured at 765 nm wavelength. Total phenolics concentration was calculated according to *Singleton & Rossi (1964)* and expressed as gallic acid equivalents, in mg/g fresh sample (FW).

Extraction crude extract of SOD and CAT: homogenized frozen tissue (2 g) with eight mL of 50 mM sodium phosphate buffer (pH 7.0), then centrifuged at 10,000 g for 10 min at 4 °C and took the supernatant. SOD activity was determined according to *Jiang et al. (2010a)* and *Jiang et al. (2010b)*. The reaction mixture contained: one mL crude extract + 750 $\mu$mol/L nitro-blue tetrazolium (NBT)+130 mmol/L methionine + 0.1 mmol/L EDTA+ 20 mmol/L riboflavin in 0.05 mol/L K-phosphate buffer (pH7.8). The reaction was started by adding riboflavin and placing the tubes of reaction mixture under 4000l×irradiance at 25 °C for 1 h. The absorbance was recorded at 560 nm. The 50% inhibition of NBT photoreduction was expressed as one SOD activity unit.

CAT activity was assayed according to the method of *Candan & Tarhan (2003)*. The reaction mixture contained: one mL of 50 mM sodium phosphate buffer (pH7.0) + one mL of 0.2% $H_2O_2$+ one mL of CAT extract. One unit of CAT activity was defined as the amount of enzyme that decomposed 1 $\mu$mol $H_2O_2$ per min at 30 °C.

POD was extracted and determined according to the methods of *Bi et al. (2011)*. Weighed 2 g of shredded and mixed mushrooms, a little quartz sand, seven mL of pre-cooled 0.1 mol/L pH5.5 sodium acetate monoacetate buffer (containing 4% polyvinylpyrrolidone), grinded on ice bath, then centrifuged (10,000 r/min 4 °C) for 30 min, and took the supernatant. The enzyme reaction system includes: three mL 25 mmol/L guaiacol + 0.5 mL the above supernatant +200 uL 0.5 mol/L $H_2O_2$ solution, quickly mixed and simultaneously recorded the absorbance at 470 nm when the reaction was 15 s, as the initial value. Record every 30 s and recorded continuously for 3 min. The absorbance value was plotted against

time, and the initial straight line portion of the reaction was taken to calculate the change value per minute. One viability unit (U) was defined as the amount of enzyme required to cause a change in absorbance of 0.01 per minute under the assay conditions.

Polyphenol oxidase (PPO) was extracted and determined according to the methods of *Li (2006)*. Weighed 2 g of shredded and mixed mushrooms, added a little quartz sand and seven mL of pre-cooled 0.05 mol/L phosphate buffer (PBS, pH5.0), grinded, then centrifuged (4 °C, 10,000 r / min, 15 min) and collected the supernatant. The enzyme reaction system includes: 2.5 mL 0.05 mol/L PBS + 0.4 mL 0.1 mol/L catechol solution (present) + 0.5 mL PPO crude enzyme solution. PPO active unit (U) was set at 0.001 change per minute at 410 nm absorbance at 20 °C.

## Analysis of cell wall metabolic enzyme activity

The activity of Chitinase and β-1,3-glucanase were extracted and determined according to the methods of *Ni et al. (2017)*. Enzyme extraction: 2 g of each sample was added seven mL of 0.1 mol/L acetic acid-sodium acetate buffer (pH5.0) and grinded, then centrifuged at 10,000 r/min for 20 min at 4 °C. The supernatants were kept at 4 °C as enzyme extractions.

Chitinase: 0.5 mL of enzyme extract was added to 0.5 mL of 50 mmol/L pH5.2 acetic acid-sodium acetate buffer and 0.5 mL of 10 g/L colloidal chitin suspension. The mixture was incubated in a water bath at 37 °C for 1 h, then, 0.1 mL of 30 g/L helicase was added and the incubation was continued at 37 °C for 1 h. After the heat preservation, immediately added 0.2 mL of 0.6 mol/L potassium tetraborate solution, 2 ml of 3,5-dinitrosalicylic acid (DNS) reagent, then boiled for 5 min. After cooling, the volume was adjusted to 25mL. The absorbance was detected at 520 nm and the boiled enzyme extraction was used as the control.

$$\text{Chitinase activity} = \frac{\text{n} - \text{acetyl glucosamine weight} \times \text{total extraction volum} \times 1000}{\text{sample volum} \times \text{reaction time} \times \text{sample weight}}.$$

β-1,3-glucanase: Two tubes were taken and one mL of 4 g/L of laminarin solution was added. Then, one mL of the enzyme solution was added to one tube, and one mL of boiled enzyme solution was added to the other tube as a control. After mixing, the reaction tube was placed in a 37 °C water bath for 40 min. After the incubation, 1.8 mL of distilled water and two mL of DNS reagent were added to the reaction tube, and boiled for 5 min, then diluted the reaction solution to 25 mL with distilled water. The absorbance of the mixture at 540 nm was measured and repeated 3 times.

$$\beta - 1, 3 - \text{glucanase activity} = \frac{\text{glucose weight} \times \text{total extraction volum} \times 1000}{\text{sample volum} \times \text{reaction time} \times \text{sample weight}}.$$

## Observation of cell structure by transmission electron microscope (TEM)

According to the methods of *Yan, Wang & Li (2002)*. The tissues of *C. comatus* with length and width of about two mm × one mm was cut out, fixed in 3.5% pentanediol, washed with 0.1 mol/L phosphate buffer of pH7.2, fixed with 1% azelaic acid, and flushed with 0.1 mol/L phosphate buffer. Then, 35%, 45%, 60%, 70%, 80%, 95%, 100% ethanol gradient

dehydration, propylene oxide transition and Spurr resin impregnation embedded. After polymerizing at 20 °C for 8 h, sliced, double stained with uranyl acetate-lead citrate, and observed by TEM.

### Statistical analysis

Data were subjected to analysis of variance (One-Way ANOVA) and least significant differences (LSD) at the 5% level ($P \leq 0.05$) were used for comparing means using the software package SPSS v24 (SPSS Inc., Chicago, USA). Graphs were created using Origin 2018 (Origin 2018., San Diego, USA).

## RESULTS

### Appearance and browning evaluation

The appearance quality of *C. comatus* during the postharvest storage was shown in Fig. 1. *C. comatus* exhibited some degree of browning, cap opening and wilting of the whole fruit body after 5 days of storage, and this situation was becoming more and more serious as the storage days went by, since store mushrooms at day 20 were the worst. After 10 days of storage, mushrooms appeared acutely browning, cap opening, stipe extending and flowing black juice, rendering the mushrooms commercially unacceptable.

The fruit body of *C. comatus* suffered from serious browning during postharvest storage either the stipe or the cap. The color difference value of cap sharply increased from 3.15 at day 0 to 27.67 at day 10 and to 57.91 at day 20, and the color difference value of stipe followed a similar trend (Fig. 2). The color difference value of the cap is basically larger than that of the stipe during storage, indicated that the cap of *C. comatus* were more susceptible to browning than stipe. Between day 5 to day 15 of postharvest storage, the $\Delta E$ of mushrooms cap increased significantly ($p < 0.05$) by 76.98%, while mushrooms stipe increased significantly ($p < 0.05$) by 91.79% between day 0 to day 5.

### Weight loss analysis and Membrane permeability

The ratio of weight loss with respect to storage time and treatments in *C. comatus* was calculated and the results showed on Fig. 3. From the Fig. 3, it is overt that on the whole there was a remarkable increase in the weight loss throughout storage, especially after 10 days weight losses were particularly excessive, thus rendering the *C. comatus* commercially unacceptable. Similar tendency in *L. edodes* was also reported by *Jiang et al. (2010a)* and *Jiang et al. (2010b)*. The weight of *C. comatus* decreased speedy, this may be due to only a thin epidermal structure on the surface of the mushroom, which does not prevent a quick superficial dehydration. *Roy, Anantheswaran & Beelman (2006)* suggested that water transpiration and $CO_2$ loss during respiration are the main causes of mushroom weight loss. While in our studies, respiration and evaporation might be the main reasons of physiological weight loss rather than transpiration because the relative humidity inside the storage chambers was near saturation.

Whether a membrane system is integrated could be reflected by membrane permeability, and the adversity stress of postharvest plants cell, which could be expressed as electrolyte leakage rate (*Ran et al., 2010*). From Fig. 3, we can visually see that the electrolyte leakage

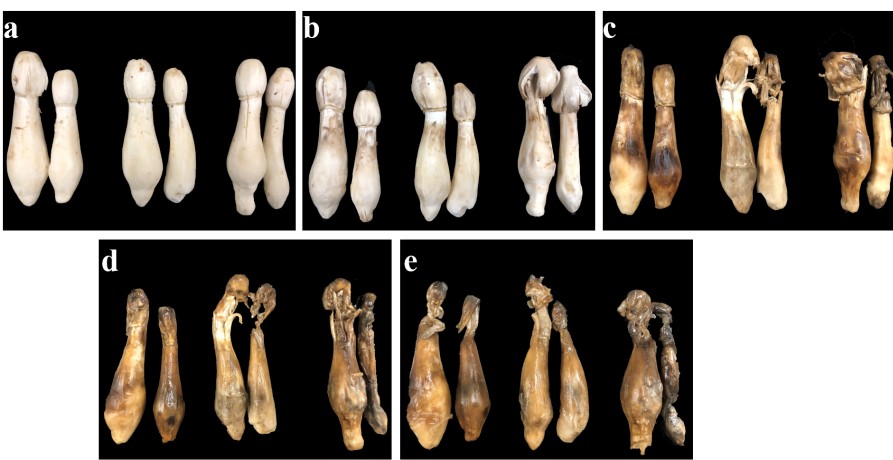

**Figure 1** **The appearance quality of *Coprinus comatus* stored at 4 °C and 90% RH.** (A) Stored for 0 d. (B) Stored for 5 d. (C) Stored for 10 d. (D) Stored for 15 d. (E) Stored for 20 d.

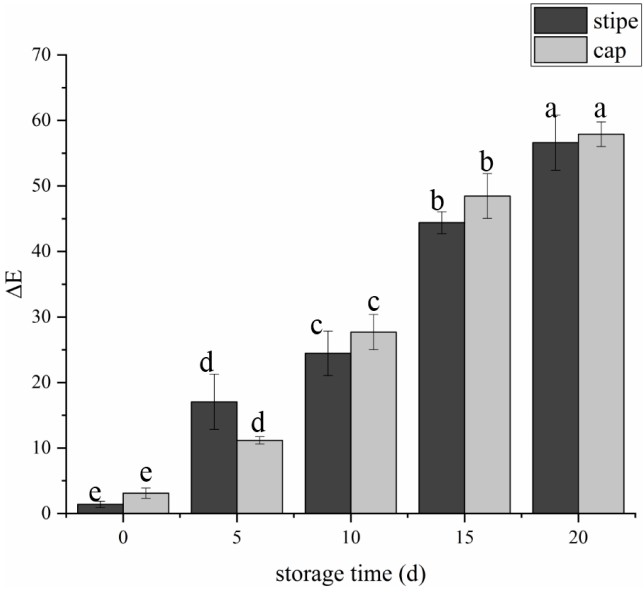

**Figure 2** **The color difference value of *Coprinus comatus* stored at 4 °C and 90% RH.**

rates of *C. comatus* rapidly increased during the whole storage (Fig. 3), reaching levels about 149.48% higher than the initial values in the end. The results indicated that *C. comatus*' membrane systems became more vulnerable to leakage during postharvest storage.

## Analysis of chemical properties

Soluble protein, as a nutrient source to support sustaining metabolic activity after picking. In our study, *C. comatus* contain approximately 3.78 mg/g FW soluble protein. A sharp decrease in soluble protein content was observed, and only 60.05% of the initial levels was

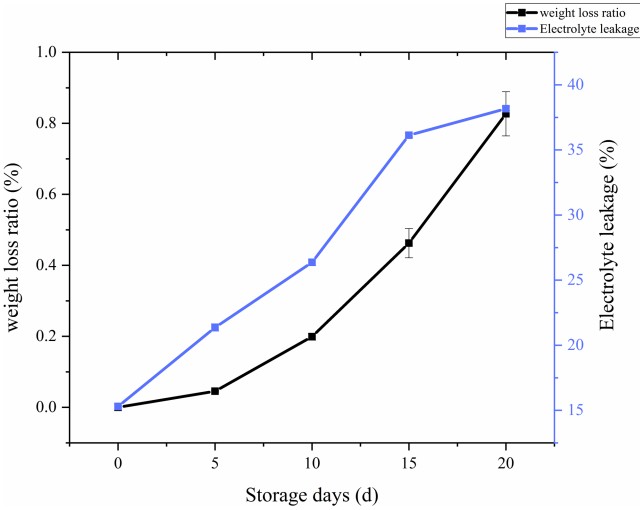

**Figure 3** The weight loss ratio and membrane permeability of *Coprinus comatus* stored at 4 °C and 90% RH.

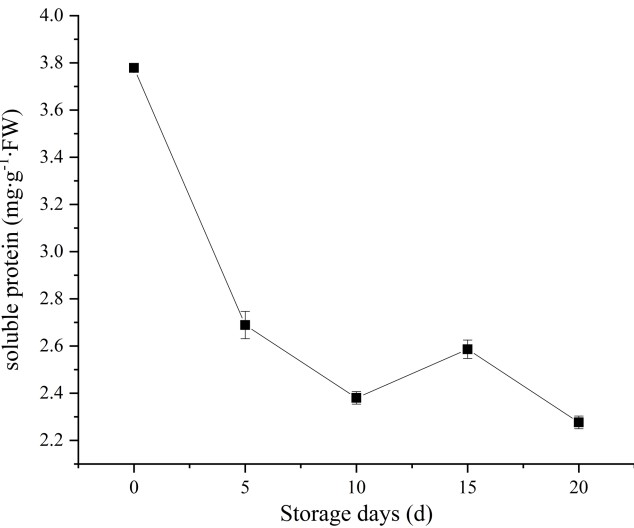

**Figure 4** The soluble protein of *Coprinus comatus* stored at 4 °C and 90% RH.

retained on the twentieth day (Fig. 4). Similar soluble protein level in *Agaricus bisporus* was also reported by *Meng et al. (2012)*. It is considered as an important indicator of tissue senescence that decline in soluble protein concentration (*Burton et al., 1997*). This decline may due to the loss of energy sources after picking.

MDA is one of the main outcomes of membrane lipid peroxidation, and its content is currently used as the indicator of membrane lipid peroxidation. It can also make large molecules such as proteins and nucleic acids become useless by altering their configuration or cross-linking (*Duan et al., 2011*). Therefore, the accumulation of MDA can damage

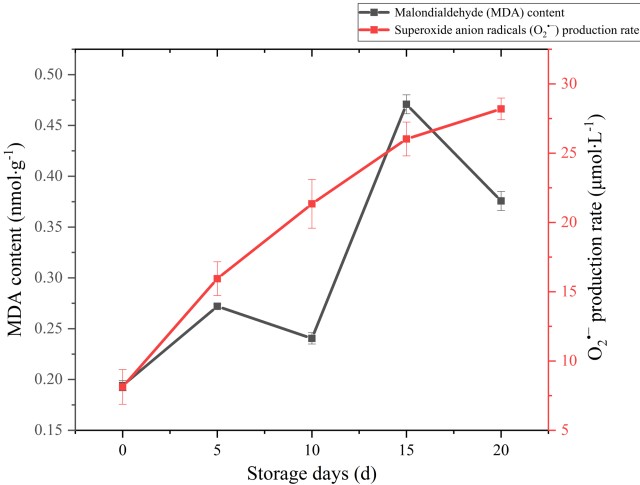

**Figure 5** The MDA content and $O_2^{\bullet-}$ production rate of *Coprinus comatus* stored at 4 °C and 90% RH.

the cytoplasm and cell membrane of mushrooms. The MDA content exhibited a gradual upward trend over the whole storage (Fig. 5), especially increased significantly ($p < 0.01$) by 95.82% between 10 and 15 days after harvest. It seemed that severely damage had happened on cell membrane after day 10 and severe membrane lipid peroxidation occurred in day 15 during the storage. MDA content had an obvious decreasing in 5–10 d. This may be related to the increase of antioxidant enzymes activity.

Reactive oxygen species (ROS) refer to several metabolites of oxygen with high chemical reactivity, such as $O_2^{\bullet-}$ and $H_2O_2$. They are considered as toxic by-products in plant metabolism, which can cause damage to macromolecular substances such as lipids (*Desikan et al., 2004*). The $O_2^{\bullet-}$ production rate sharply augmented throughout the postharvest period from 8.13 umol/L to 28.19 umol/L (Fig. 5), which increased significantly ($p < 0.01$) by 246.74%. The growth of $O_2^{\bullet-}$ production rate was obvious in the first five days, slowed down in the after 15 days during storage.

## Analysis of functional components

SOD, CAT and POD are the main antioxidant enzymes in edible fungi. During the mushroom postharvest ripening process, they played a vital role in antioxidant defense and were thought to extend food shelf life by protecting the integrity of membranes (*Gill & Tuteja, 2010*; *Xing et al., 2007*). Time courses of SOD activity is shown in Fig. 6. SOD activity increased slowly at the beginning of postharvest storage, afterward increased rapidly. peaked at the day 10 (31.62 U g$^{-1}$ Fw), and then exhibited a sharply downward trend. A CAT activity increase was watched during the first 15 days of storage, and then exhibited a slight drop over the next 5 days (Fig. 6). A relatively large increase of CAT activity was recorded between day 5 to day 15, with an increase of 30 folds. POD could help plants get rid of reactive oxygen species (ROS) and was one of the membrane lipid peroxidation defense systems of fruit and vegetable (*Terefe, Buckow & Versteeg, 2014*). The changes of POD activity during the postharvest storage are also displayed in Fig. 6. A rapid

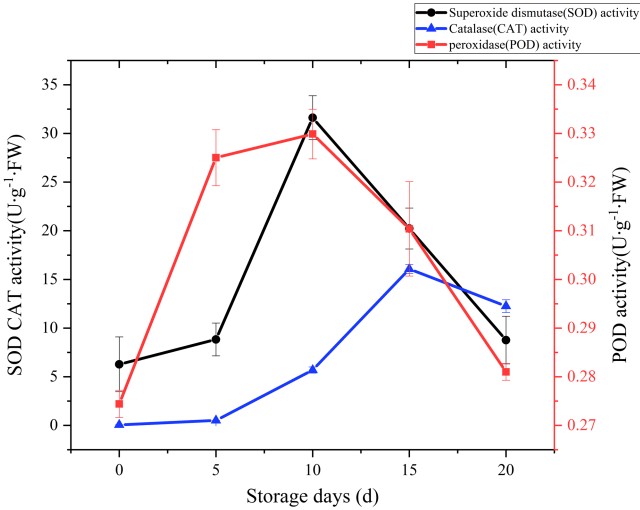

**Figure 6** **The SOD, CAT, POD activity of *Coprinus comatus* stored at 4 °C and 90% RH.**

increase was observed during the first 5 days, then slowly up to a higher peak (0.33 U/g FW) at day 10, afterwards sharply dropped to raw value.

A sharply increase was observed in 10 days after harvest, with a higher peak of total phenolics (0.89 mg g$^{-1}$ FW). Then total phenolics level showed a gradual downward trend over the next 10 days (Fig. 7). The oxidation of phenolic substances was the main factor of browning in mushroom tissue, which is one of the key factors that influence the shelf-life and product quality of the postharvest mushrooms. In our case, the content of total phenols increased significantly ($p < 0.01$) by 13.39% between 5 and 10 days after harvest, it may be caused by the reduction of PPO activity, and the occurrence and timing of peak roughly coincides with the cap browning of *C. comatus*.

The change of PPO activity of *C. comatus* during postharvest storage is given in Fig. 7. It increased rapidly from the initial value, then decreased, afterwards increased to a peak, and finally decrease, nonetheless on the whole there was an uplifted tendency. Similar tendency in *Agrocybe aegirit* was also reported by *Lo & Cheung (2005)*. PPO participation in enzymatic browning was thought to be the major factor of discoloration of many food (*Vámos-Vigyázo & Haard, 1981*). It catalyzed the oxidation of polyphenols into quinones, which turned into melanin by polymerization, thus severely affecting the mushrooms' nutritional value, flavor and appearance quality. Hence, in order to preventing the synthesis of melanin in the browning of mushrooms, control of PPO activity is indispensable. Increased PPO activity was associated with mushroom browning, suggesting that the browning of *C. comatus* after postharvest was possibly attributed to the action of PPO.

## Cell wall metabolism-related enzyme activity

The change of cell wall metabolizing enzyme activity of *C. comatus* are given in Fig. 8. Studies have shown that the content of chitin in edible fungi decreases due to the effect of chitinase in the storage process. Chitinase hydrolyzes the beta-1, 4-glycoside bond in

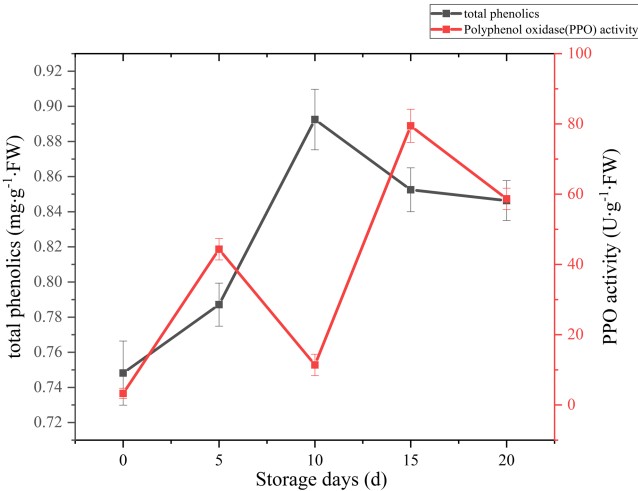

**Figure 7** The total phenolics and PPO activity of *Coprinus comatus* stored at 4 °C and 90% RH.

chitinol to produce n-acetylglucosamine oligomer or monomer (*Buitimea et al., 2013*). *Lim & Choi (2009)* found that the *Agaricus bisporus* could produce black juice in the process of autolysis at 25 °C for 15 h. By PCR confirmed that the amount of chitinase synthesis in the process of autolysis of *A. bisporus mushroom* was significantly higher than that in the normal *A. bisporus mushroom*. In this study, the activity of Chitinase increased rapidly from the initial low value to a peak during the first 10 days, and finally declined over the next 10 days. β-Glucanase is a cell-wall-degrading enzyme mainly effective on glucans. β-1,3-glucanase activity of *C. comatus* increased up to day 10 of storage and decreased thereafter. Thus, the higher degrees of cell wall polysaccharide degradation observed during storage might be due to their high activities.

## Changes in stipe ultrastructure

At 5,000 times magnification, we observed the cell of fresh *C. comatus* maintained complete structure as shown in Fig. 9A. Cells arranged in neat rows and the cell wall structure was complete. There are many nucleus and mitochondria in the cytoplasm. Cell of *C. comatus* stored for 10 d was shown in Fig. 9B, the cells no longer line up. The distance between cells increased and the thickness of the cell wall decreased, at the same time, the number of nucleus and mitochondria decreased. Ultrastructure of *C. comatus* stored for 20 d was given in Fig. 9C, which showed that cells had lost integrity already. Cell wall cracked and cell contents permeated outside. Hence, *C. comatus* value had been lost.

At 10,000 times magnification, an integrated cell was observed in Fig. 9D, the nucleus and organelles were clearly visible. A relatively integrated cell was observed in Fig. 9E, the nucleus and the organelles become blurred. A dissolved cell was observed as shown in Fig. 9F, the nucleus and organelles were completely degraded into a dark mass.

At 15,000 times magnification, the structure of mitochondria was clear, and mitochondrial cristae could be observed in fresh *C. comatus* (Fig. 9G). Cell of *C. comatus* stored for 10 d was shown in Fig. 9H, mitochondria still existed and mitochondrial

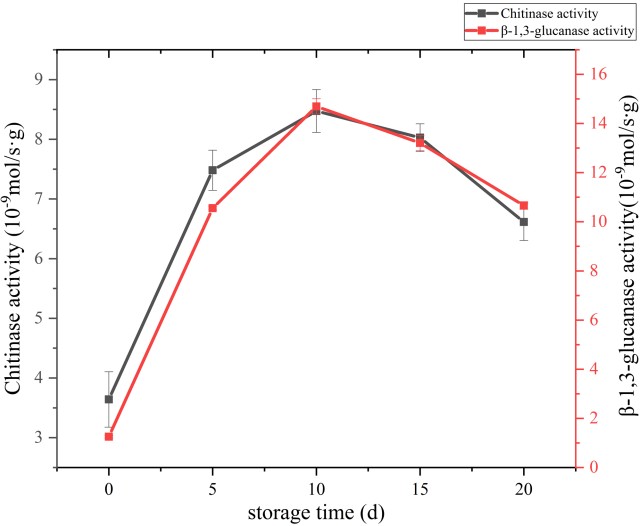

**Figure 8** **The cell wall metabolizing enzyme activity of *Coprinus comatus* stored at 4 °C and 90% RH.**

membrane was visible. While, the mitochondria had almost degraded when stored for 20 d (Fig. 9I), which just reserved the structure of mitochondrial was disabled.

## DISCUSSION

In this study, we created a *C. comatus* softening model to evaluate changes of sensory characteristics, chemical constituents, antioxidant activities and secondary metabolism of the cell wall during a 20-day storage. *C. comatus* exhibited some degree of browning, wilting of the whole fruit body and cap opening after 5 days of storage, and this situation was becoming more and more serious as the storage days went by. The cap of *C. comatus* were more susceptible to browning than stipe. Due to removing waters and nutrients after harvest, weight loss and soluble protein declined. The continuous production of MDA and $O_2^{\bullet-}$ leads to the increasing permeability of cell membrane, further leads to membrane lipid peroxidation. While, the increased activity of SOD, CAT and POD can beneficial in protecting cells from ROS-induced injuries, alleviating lipid peroxidation and stabilizing membrane integrity.

Color and textural changes are the two main factors limiting mushroom quality and shelf life (*Soler-Rivas et al., 2010*). In our study, relatively high positive correlation ($r \geq 0.789$) was found between browning and polyphenol oxidase (PPO) enzyme. Thus, the browning of *C. comatus* after postharvest was possibly attributed to the action of PPO. May be by inhibiting the activity of PPO, the color quality of *C. comatus* can be effectively improved. Firmness decrease during storage is generally attributed to changes in cell wall composition (*Manolopoulou et al., 2007*; *Singh et al., 2010*). Ripening-related modifications in cell wall composition and structure are commonly attributed to the finely tuned, coordinated action of a number of specific enzymatic and non-enzymatic proteins on different cell wall polysaccharides, which eventually lead to cell wall disassembly (*Belge et al., 2015*). *Ni*

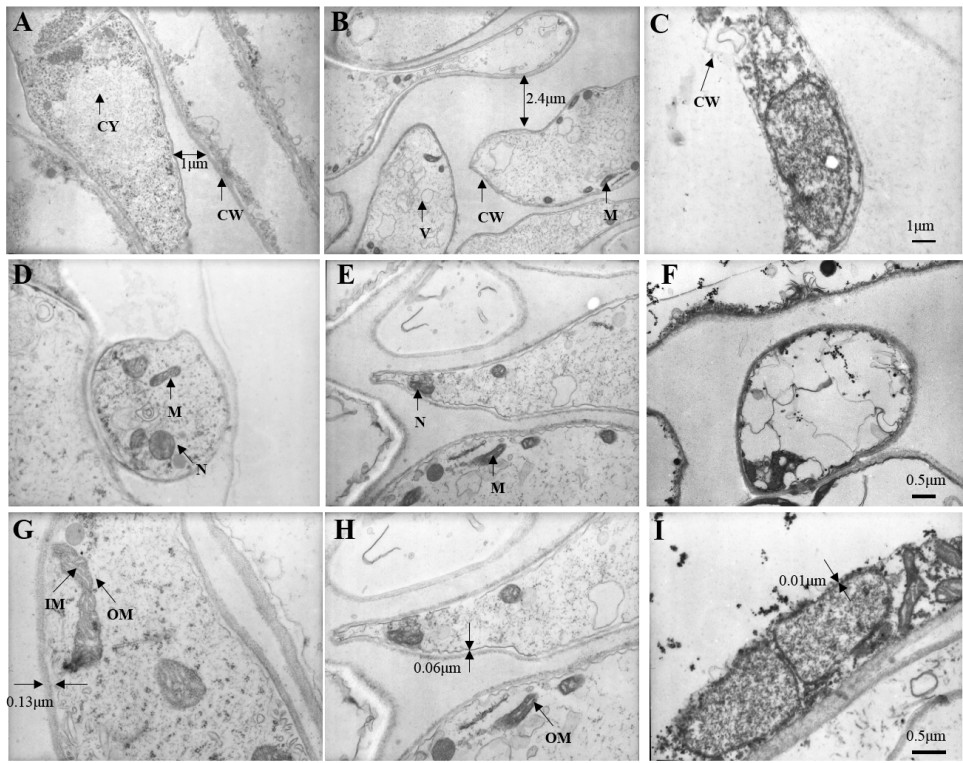

**Figure 9  Changes in stipe ultrastructure of *Coprinus comatu s* though TEM stored at 4 °C and 90% RH.** (A) Storage for 0 d, (B) Storage for 10 d, (C) Storage for 20 d. amplification: 5 ×1000. (D) Storage for 0 d, (E) Storage for 10 d, (F) Storage for 20 d. amplification:10 ×1000. (G) Storage for 0 d, (H) Storage for 10 d, (I) Storage for 20 d. amplification:15 ×1000. CW, cell wall; CY, cytoplasm; M, mitochondria; N, nucleus; V, vacuole; IM, mitochondrial inner membrane; OM, mitochondrial outer membrane.

*et al. (2017)* studied secondary metabolism associated with softening of *Lentinula edodes* at room temperature, and found that glucanase activity reached the highest level at day 3, chitinase activity and cellulase activity increased since day 3 still day 6 during storage. *Karakurt & Toka (2016)* studied the effects of $CaCl_2$ and hot water dips on the change hydrolase activities of *Agaricus bisporus*, and found xyloglucanase and beta glucanase activities increased significantly during storage. In our study, the cell wall metabolizing enzymes, including Chitinase, β-1,3-glucanase, they all increased firstly and reached the highest level at day 10 during postharvest storage, then decreased. Thus, it is possible that the higher degrees of cell wall degradation observed during storage might be due to their high activities. *Liu et al. (2015)* suggested that β-1,3-glucanases plays a major role in the autolysis of *Coprinopsis cinerea*. In our experiment, the activity of β-1,3-glucanases was the strongest among cell wall metabolizing enzymes. It is also known that β-1,3-glucan with β-1,6-glucan braches are the backbone of the fungal cell wall network (*Aimanianda et al., 2017*). Based on the above, we speculate β-glucanases could destroy the cytoskeleton of *C. comatus*, while the chitinases may degrade cell walls in synergy with β-1,3-glucanases. Surely, the activity of enzymes is ultimately regulated by genes expression, so more experiments are needed to verify this speculation in the next time. The ultrastructure changes of the postharvest *C.*

*comatus* were observed for the first time. The nucleus, mitochondria had disappeared and thinner cell wall were detected with the storage duration. The lower integrity of cell walls may result in greater loss of protein and polysaccharide.

ROS accumulates and leads to lipid peroxidation when plants suffer from biotic and abiotic stresses (*Ren et al., 2012*). MDA is a lipid peroxidized product, and it can reflect the extent of ROS-induced membrane lipid peroxidation (*You et al., 2012*). In this experiment, the MDA content exhibited a gradual upward trend over the whole storage, especially increased significantly ($p < 0.01$) by 95.82% between 10 and 15 days after harvest. The result seems to indicate that severely damage had happened on cell membrane after day 10 and severe membrane lipid peroxidation occurred in day 15 during the storage. To protect cells from ROS-induced injuries, plant tissues generated SOD, CAT and POD to scavenge ROS (*Duan et al., 2011*). In our experiment, showed high activity and the $O_2^{\bullet -}$ production rate increased slowly between 5 and 15 days after harvest. It validates that high SOD, CAT, POD activity could be beneficial in protecting cells from ROS-induced injuries, alleviating lipid peroxidation and stabilizing membrane integrity. Although the activity of SOD, CAT, POD of *C. comatus* peaked up to 31.62 U g$^{-1}$ Fw, 16.51 U g$^{-1}$ Fw, 0.33 U g$^{-1}$ FW, respectively, they were lower than that of *Agaricus bisporus* (*Li et al., 2019*) at the similar storage conditions. This indicated that enzymatic antioxidant defense of *C. comatus* was inferior to that of *Agaricus bisporus*. It also means *C. comatus* were more likely to spoil after harvest. *Ding et al. (2016)* studied the effects of postharvest brassinolide (BL) treatment on the metabolism of *Agaricus bisporus*, and found BL treatment significantly decreased the accumulation of ROS and induces the antioxidant enzyme system. *Li et al. (2019)* also studied storage quality of *Agaricus bisporus* and the result showed that 10 mM L-arginine treatment inhibited PPO activities, while inducing SOD and POD activities. *Lia et al. (2013)* explored low oxygen and high carbon dioxide storage effects on reactive oxygen species metabolism in *Pleurotus eryngii*. They found the activities of SOD, POD, and CAT in 2% $O_2$ + 30% $CO_2$ treated mushrooms were significantly higher than those of the control (ambient air). In a study by *Shi et al. (2017)*, the nanocomposite packaging material could effectively delay the postharvest senescence of *Flammulina velutipes*. *Jiang et al. (2010a)* and *Jiang et al. (2010b)* reported four macroholes treatment was a useful way of maintaining *Lentinula edodes* texture during storage at 4 °C. Above all, it is possible to apply these methods to the storage of *C. comatus*, hence prolonged its shelf life. Furthermore, in the future, the expression of genes which associated with antioxidant activity and cell wall metabolism could be artificially regulated through gene editing, so as to extend the life of *C. comatus* and increased its preservation quality.

## CONCLUSIONS

To sum up, these results provide insights into the quality deterioration of *C. comatus* after postharvest. Identifying and reducing the quality deterioration is necessary to provide the edible fungus industry more appropriate recommendations for postharvest management, which would ultimately maximize both sensory and nutritional quality maintenance of fresh mushrooms.

### Funding

This work was financially supported by the Technology of Sichuan Province (No. 2016NFP0091) and the National Natural Science Foundation (No. 31870309). The funders had no role in study design, data collection and analysis, decision to publish, or preparation of the manuscript.

### Grant Disclosures

The following grant information was disclosed by the authors:
Technology of Sichuan Province: 2016NFP0091.
National Natural Science Foundation: 31870309.

### Competing Interests

The authors declare there are no competing interests.

### Author Contributions

- Yi Peng conceived and designed the experiments, performed the experiments, analyzed the data, prepared figures and/or tables, authored or reviewed drafts of the paper, and approved the final draft.
- Tongling Li, Cairong Yang and Wei liang Qi performed the experiments, prepared figures and/or tables, and approved the final draft.
- Huaming Jiang, Yunfu Gu and Qiang Chen analyzed the data, prepared figures and/or tables, and approved the final draft.
- Song-qing Liu and Xiaoping Zhang conceived and designed the experiments, authored or reviewed drafts of the paper, and approved the final draft.

### Data Availability

Data is available in the Supplemental Files.

### Supplemental Information

Supplemental information for this article can be found online at http://dx.doi.org/10.7717/peerj.8508#supplemental-information.

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
