# Peer review of "Postharvest biochemical characteristics and ultrastructure of Coprinus comatus"

_PeerJ, doi:10.7717/peerj.8508_

## Round 0.1 · original submission · Major Revisions

We have received four reports with detailed suggestions for the improvement of your paper. I look forward to receiving an updated version!

Reviewer 1 ·

Basic reporting

In this paper, the change of color, SOD, CAT and POD activities, chitinase, β-1,3-glucanase, Cx and PG activities and the ultrastructure of Coprinus comatus was analyzed. The research does have some significance to understand why C. comatus is difficult to keep fresh. However, the paper should be major revised before being published.

Major comments

This paper shown the postharvest change of C. comatus, but did not give any suggestion about how to extend the storage life of C. comatus. Author declared that, by inhibiting the activity of PPO, the color quality of C. comatus can be effectively improved. This should be proved by experiment. At last, the author should change the storage condition to decease the postharvest change of C. comatus. Meanwhile, in the discussion, possible technologies to extend the storage life of C. comatus should be discussed.

Writing is very poor. The ms should be revised carefully and checked by native speakers in scientific American or British English. A lot of discussion was in the results part of ms. The results should be describe more clear.

In the ultrastructure changes, the organelles, nucleus, mitochondria et al., should be marked in the Fig., and the thickness of cell wall should be measured and counted.

Line 149. P value for statistics is incorrect. Normally, p < 0.05 means significant differences. The method for calculating the p value should be described in method. The significant differences also should be marked in the figures.

Minor comments

C.comatus should be C. comatus.

Line 50. C. comatus should in Italics

Line 68-70. Lim et al. (Lim et al., 2009) should be Lim et al. (2019). Other references also need to be revised according the requirements of the journal.

Figure 5. “umol/L” should be “mol/L” in symbol typeface.

Fig. 2 was not cited in the ms. Fig. 1 and Fig. 2 could be merged.

Every abbreviation should be shown first with its full form. Such as “FW”, “MDA” and so on in the ms.

Line 231, et al. bisporus mushroom is not the Latin scientific name, and should be agaricus bisporus.

Experimental design

The detail of method should be provided even it is reported by other paper.

Validity of the findings

no comment

Additional comments

no comment

Reviewer 2 ·

Basic reporting

This manuscript is described about "Postharvest biochemical characteristics and ultrastructure of Coprinus comatus". It is good manuscript for the readers of PeerJ.
I have some recommendations about this manuscript.

1. You should quote a little more papers in the section of "Discussion".
Line 274-277 : Are these activities of SOD, CAT, POD higher or lower than another mushrooms.

Line 287-289: Do the activities of chitinase, Cx, PG, beta-1,3 glucanase increase during postharvest storage in other mushrooms.

2. About the localization of chitinase, Cx, PG, beta-1,3-glucanase
You didn't discuss about the localization of chitinase, Cx, PG, beta-1,3-glucanase (autolysis enzymes). You should discuss about it.

Experimental design

It is ok about experimental design.

Validity of the findings

This manuscript is good findings about postharvest biochemical characterization and ultrastructure of C. comatus.

Reviewer 3 ·

Basic reporting

no comment

Experimental design

no comment

Validity of the findings

no comment

Additional comments

This manuscript reported postharvest biochemical characteristics and ultrastructure of Coprinus comatus and showed some interesting results, however there are some mistakes in the scientific names of the mushroom strain in this manuscript and some concepts of chemical components and structures of fungal cell walls. Therefore, the supplemented data and major revision are needed for further review.

1. Line 70-77. By study of a autolysis mutant of C. cinerea fruiting bodies explored that β-1,3-glucanases plays a major role in the autolysis of Coprinopsis fruiting bodies (J. Agric. Food Chem. 2015, 63, 9609−9614). Authors should cite this recent progress.

2. Line 92. Was the C. comatus strain used in this study really characterized as a C. comatus stain by morphological identification and molecular identification? I felt that the fruiting bodies shown in the manuscript were very different from common C. comatus fruiting bodies.

3. Line 100-104. color difference meter, producer? What do L, a, b, E, and

·

Basic reporting

The article shows some novel informations about the Coprinus comatus mushroom activity during storage. The English is professional, just some minor modifications might improve the text quality (for example: row: 36-38; 153-158).
The cited references are correct, and contain all the necessary articles. The charts and figures and clear, professional. I couldn't see the raw data, but the output of those can be adequate.

Experimental design

The authors worked with traditional methods and used common techniques for the research. However the used methodology are clear and might be some evidence, the description of those is completly missing. The materials and methods chapter are citing the references, but the way of measurement is missing (line: 106, 108, 110-113, 115-120, 123-126). In my point of view, the proper description of the used methodology (and not the reference's description) is highly improves the acceptance of the results. The instruments, chemicals and details about measurements are needed.

Validity of the findings

The output of the results are claer, and confirmed by the measurements. The conclusions are supported by the charts and are well stated.

Additional comments

- The cultivation technology of the mushrooms have a high influence onto the shelf life and storage. The irrigation during the crop, the compost (is it a compost or a fermented subtrate, sterilised/pasteurised) and the environmental conditions (CO2, RH%, temperature) are having influence into the mushrooms picked. Therefore, some technical data about crop management would improve the text and helps to understand the chemical changes during storage.
- The Fig. 1 shows the Coprinus mushrooms. They are young and looks no light during cultivation were used. Does it have any influence onto the mushroom's quality? The proper identification of the mushroom was done by classical methods?
- In the text a phrase 'Bisporus mushroom' (line: 68, 231) is mentioned but not clear what does it mean. Is it refer to Agaricus bisporus?
- A list of abbreviations would nice to have (PPO, MDA, etc) and the full name would great to write into the title of the figures (Fig. 5,6, 7)

---

## Round 0.2 · Minor Revisions

I agree with the reviewer regarding your mentions of a possible action of cellulase on the fungus cell wall. Please rephrase or replace them.

Reviewer 3 ·

Basic reporting

no comment

Experimental design

no comment

Validity of the findings

no comment

Additional comments

The revised manuscript still has some issues need to be modified further for publication in the journal.

1. Authors indicate that the C. comatus strain used in this study is a domesticated cultivar of C. comatus77 which looks different from the wild. Authors should give the reference for the strain in which the shape of fruiting bodies was shown and readers can know its difference from common C. comatus fruiting bodies. Otherwise, authors need to give identification data.

2. The revised manuscript still lacks the information of the company for production of color difference meter.

3. Authors have deleted the description of mushroom cell walls containing cellulose and pectin in the revised manuscript. However, the answers to comments are not good.
It is a scientific knowledge that fungal cell walls do not contain cellulose and pectin as components unless new experimental evidences are presented in the future.
As described by Zhen (2011), cellulose in tissues was digested and washed out with H2SO4, the difference of weights between the acid-treated tissue and the non-treated tissue represented the cellulose. Apparently, here the method for analysis of plant cell walls was incorrectly used for analysis of fungal cell walls. In plant cell walls, cellulose is insoluble whereas hemicellulose is soluble. So cellulose could be hydrolyzed with and washed out with H2SO4 to determine the cellulose content in the plant cell walls. However, in fungal cell walls such as basidiomycete cell walls, insoluble components are chitin and beta-1,6-branched beta-1,3-glucans. So, Zhen (2011)'s document did not provide a convinced evidence for existing of cellulose in mushroom cell walls. So far, no definitely report proves that basidiomycetes cell walls contain cellulose except some reports from Chinese scholars. In fact, fungal cell walls contain some beta-1,3/1,4-glucans which is different from cellulose (J Biol Chem, 2000, 275(36): 27594–27607; Molecular Microbiology, 2007, 66(2): 279–290). However, cellulases did not hydrolyze it (FEBS Letters, 52(2): 202–207).
According the methods for analysis of pectin content described by Shen et al (2013), so-called pectin was extracted from mushroom fruiting bodies with boiled water or boiled acid solution. Again, here the method for analysis of plant cell walls was incorrectly used for analysis of fungal cell walls. No scientist characterized a component as a pectin in fungal cell walls so far. We can image that some components could be extracted from mushroom in Shen's experiment but they were not pectin. So, we could not get conclusion that fungal cell walls contain pectin component only according to the non-specific extraction and quantitative analysis of pectin in fungal cell walls.

4. References format are chaos and must be modified carefully.

---

## Round 0.3 · accepted · Accept

Thank you for performing the last required corrections.